# Environment and Behavior: Neurochemical Effects of Different Diets in the Calf Brain

**DOI:** 10.3390/ani9060358

**Published:** 2019-06-14

**Authors:** Angelo Peli, Annamaria Grandis, Marco Tassinari, Paolo Famigli Bergamini, Claudio Tagliavia, Mariana Roccaro, Cristiano Bombardi

**Affiliations:** Department of Veterinary Medical Sciences, University of Bologna, Via Tolara di Sopra, 50, 40064 Ozzano dell’Emilia (BO), Italy; angelo.peli@unibo.it (A.P.); annamaria.grandis@unibo.it; (A.G.) marco.tassinari@unibo.it (M.T.); paolo.famigli@unibo.it (P.F.B.); claudio.tagliavia2@unibo.it (C.T.); mariana.roccaro2@unibo.it (M.R.)

**Keywords:** prefrontal cortex, paraventricular nucleus, nucleus of the solitary tract, calf, diets

## Abstract

**Simple Summary:**

Social stress is characterized by physiological changes in brain functions as well as changes in neuroendocrine system activities. Much evidence indicates that stress responses are mediated by three main stress-responsive cerebral regions: the prefrontal cortex, the paraventricular nucleus of the hypothalamus, and the nucleus of the solitary tract of the brainstem. This is the first study investigating the effects of different diets on the prefrontal cortex, the paraventricular nucleus, and the nucleus of the solitary tract in ruminants. The results obtained suggest that weaning not only reduces the stress and, consequently, alterations in the oxytocin neuronal population of the paraventricular nucleus but also stimulates myelination in the prefrontal cortex. These data support the importance of maintaining a healthy nutritional lifestyle, similar to that occurring in natural conditions.

**Abstract:**

Calves reared for the production of white veal are subjected to stressful events due to the type of liquid diet they receive. Stress responses are mediated by three main stress-responsive cerebral regions: the prefrontal cortex, the paraventricular nucleus of the hypothalamus, and the nucleus of the solitary tract of the brainstem. In the present study, we have investigated the effects of different diets on these brain regions of ruminants using immunohistochemical methods. In this study, 15 calves were used and kept in group housing systems of five calves each. They were fed with three different diets: a control diet, a milk diet, and a weaned diet. Brain sections were immunostained to evaluate the distribution of neuronal nitric oxide synthase and myelin oligodendrocyte glycoprotein immunoreactivity in the prefrontal cortex; the expression of oxytocin in the paraventricular nucleus; and the presence of c-Fos in the A2 group of the nucleus of the solitary tract. The main results obtained indicate that in weaned diet group the oxytocin activity is lower than in control diet and milk diet groups. In addition, weaning appears to stimulate myelination in the prefrontal cortex. In summary, this study supports the importance of maintaining a nutritional lifestyle similar to that occurring in natural conditions.

## 1. Introduction

Calves reared for the production of white veal are fed with a predominantly liquid diet composed of milk-replacer and, in accordance with European law (Council Directive 2008/119/EC laying down minimum standards for the protection of calves), some fibrous feed, increasing from 50 to a minimum of 250 g/d from the beginning to the end of the fattening cycle, at 20–26 weeks of age. Thus, these animals are never truly weaned and, although they are ruminants, they remain functionally monogastric. Even though normal prestomachal development does not occur, white veal calves still try to ruminate, but cannot show the full rumination behavior since they have received insufficient solid material in their diet. The lack of chewing and ruminating opportunity is thought to cause frustration, chronic stress and hence poor welfare, which is expressed by abnormal oral behaviors [1].

Social stress is characterized by physiological changes in brain functions as well as changes in neuroendocrine system activities [2]. Much evidence indicates that stress responses are mediated by brain areas located in the cortex regions, the hypothalamus, and the brainstem [3]. In particular, three main stress-responsive regions can be identified in the brain: the prefrontal cortex, the paraventricular nucleus of the hypothalamus, and the nucleus of the solitary tract of the brainstem.

The prefrontal cortex, a six-layer isocortex, is responsible for the efficient operation of a number of cognitive functions, such as attention, memory, and behavioral flexibility [4]. In addition, this area is thought to mediate response to stress. Although the precise mechanisms involved remain unclear, a recent study indicates that neurotransmission mediated by nitric oxide is increased in the prefrontal cortex with long-term anxiogenic effects [5]. In addition, protracted social isolation of adult mice impairs myelination in their prefrontal cortex [6].

Sensory feedback signals from the body are conveyed to widely distributed regions of the brain, including the paraventricular nucleus and the nucleus of the solitary tract [7]. The paraventricular nucleus is a small set of neurons residing in the medial zone of the hypothalamus [8,9]. In the paraventricular nucleus, magnocellular and parvocellular neurons can be identified and classified on the basis of their neurochemical profile [8,9]. Generally, both magnocellular and parvocellular neurons can synthesize oxytocin [10]. Magnocellular neurons, through the hypothalamo-neurohypophysal system, release the oxytocin directly into the systemic circulation, modulating different physiological activities such as labor, parturition, lactation, maternal behavior, appetite regulation, gastric distension, blood pressure, and electrolyte balance [11]. In addition, these cells, releasing the oxytocin into portal circulation, modulate adrenocorticotropic hormone (ACTH) release, and should be considered possible modulators of stress response [12,13,14]. Parvocellular neurons that secrete oxytocin, on the other hand, projecting to brainstem and spinal cord autonomic neurons, integrate sympathetic/parasympathetic outflow and regulate the activity of the adrenal cortex through an autonomic system [11]. Stress induces profound changes in the neuroendocrine system. Interestingly, in caged mice, levels of anxiety and depression induced by chronic restraint stress can be reduced through the elevation of oxytocin-immunoreactive (IR) neurons in the paraventricular nucleus [2]. In addition, corticotrophin realizing factor 2 receptors located in the oxytocinergic neurons of the paraventricular nucleus can regulate the release of the oxytocin during times of stress [15].

The nucleus of the solitary tract is located in the dorsal medulla and is involved in many integrative functions including autonomic control and neuroendocrine processes [7,16]. Neurochemically, the intermediate part of the nucleus of the solitary tract contains noradrenergic neurons (A2 group) that provide numerous projections to the paraventricular nucleus [10]. These cells are activated by a variety of acute and chronic stressors [17,18,19] and receive descending inputs from the prefrontal cortex. Interestingly, stressor signals are able to induce c-Fos (a marker of neuronal activity) expression in the nucleus of solitary tract neurons [17,20,21].

Many studies suggest that a variety of stressors affect the activity of different circuits located in the forebrain, the hypothalamus, and the brainstem of rodents [2,5,20,22,23,24]. However, nothing is known about the effects that stress related to the intake of different diets can induce on brain structures and the endocrine system of bovines.

In the present study, we addressed this knowledge gap by using immunohistochemical techniques to evaluate the distribution of neuronal nitric oxide synthase (nNOS) and myelin oligodendrocyte glycoprotein (MOG) immunoreactivity in the prefrontal cortex, as well as the expression of oxytocin in the paraventricular nucleus; the presence of c-Fos in the A2 group of the nucleus of the solitary tract. The lack of provision of solid feed in adequate amounts, which allows for proper rumen development and normal rumination activity, is one of the main welfare problems for intensively reared calves [25]. It is important to note that the possibility of expressing normal behavior and of satisfying needs in terms of quantity as well as quality of feed are two of the Farm Animal Welfare Council’s Five Freedoms [26].

## 2. Materials and Methods

### 2.1. Animals and Ethical Statement

The study was carried out from April to September 2014 in a commercial white veal calves’ facility, in the North of Italy. All animal procedures performed in this study met the requirements of Italian law on the use of animals for experimental and other scientific purposes (Legislative Decree 26/2014, implementing Directive 2010/63/EU on the protection of animals used for scientific purposes) and the research protocol was approved by the Ethical-Scientific Committee of the University of Bologna (n°28/79/2014). The legislation in force in the European Union (EU) (Council Directive 2008/119/EC) lays down the minimum standards for the protection of calves, including provisions on an appropriate diet adapted to their age, weight, and behavioral and physiological needs, to promote good health and welfare. Calves must be fed according to their physiological needs and, in particular, fiber food must be provided to ensure proper rumen development and, subsequently, effective rumination activity which is, undoubtedly, a typical behavioral need of ruminants. 

In this study, 15 Holstein Friesian intact male calves (15–25 days old) were used. They were standard housed over a six-month period, until slaughtering. All calves were kept in a group housing system of five calves per pen (a small enclosure for livestock) and they were fed with three different diets: the control diet group was fed with a commercial diet commonly used for white veal calves, in accordance with EU law, composed of milk-replacer (two daily meals increasing from 2.5 L/meal/calf to 8.5 L/meal/calf at the end of the fattening cycle) and the concentrate as a fibrous source (gradually increasing from 50 g/calf to 1500 g/calf daily). The milk diet group was fed only with milk-replacer, in the same quantities as group 1. The wean diet group was fed in the same manner as the control diet group, with a supplement of hay, until 60 days after arrival at the fattening unit, when provision of milk-replacer was stopped, and rearing continued on a solid feed diet of concentrate and hay. At the beginning of the research, the average weight of the calves was as follows: 49.40 kg for the control diet group; 49.60 kg for the milk diet group; 49.91 kg for the wean diet group. At the end of the housing period, after 183 days, the average weight of the animals was 278.88 kg for the control diet group, 232.38 kg for the milk diet group and 213.88 kg for the wean diet group.

### 2.2. Fixation and Sample Collection

The brain was removed at the slaughterhouse, where animals were treated according to the European Community Council Regulation (CE/1099/2009) concerning animal welfare during the commercial slaughtering process and were constantly monitored under mandatory official veterinary medical care. All the animals were considered free of pathologies by the veterinary medical officer responsible for the health and hygiene of the slaughterhouse. 

The gyrus proreus of the left and right frontal lobes, including the prefrontal cortex, the hypothalamus, and the brainstem, were fixed by immersion in 4% paraformaldehyde 0.1 M sodium phosphate buffer, pH 7.4 for 72 h. After rinsing in phosphate buffer saline (PBS, pH 7.4) the tissue was cryoprotected in 30% sucrose solution in PBS (pH 7.4) at +4 °C and cut on a sliding freezing microtome in serial coronal sections (50 μm). The sections (1 in 5 series) were stored in 30% sucrose solution in PBS at −20 °C (for immunohistochemical staining) or in 10% formalin at room temperature (for thionin staining, Sigma, Saint Lous, MO, USA) until processed.

### 2.3. Immunoperoxidase Experiments

The free-floating coronal sections were washed in 0.02 M PBS, pH 7.4 (three times for 10 min each). To eliminate endogenous peroxidase activity, the sections were treated with 1% H_2_O_2_ in H_2_O for 15–30 min at room temperature. Sections were rinsed in 0.02 PBS (six times for 5 min each) and incubated in a solution containing 10% normal goat serum (Colorado Serum Co., Denver, CO, USA, #CS 0922) and 0.5% Triton X-100 in 0.02MPBS for 2 h at room temperature to block non-specific binding. Thereafter, the sections were incubated in a solution containing rabbit anti-oxytocin polyclonal antibody (diluted 1:100; code PA5-26701; ThermoFisher Scientific, Bannockburn, IL, USA) or rabbit anti-MOG polyclonal antibody (diluted 1:100; code PA5-19602; ThermoFisher Scientific, Bannockburn, IL, USA) or mouse anti-nNOS monoclonal antibody (dilute 1:500; code TA300971; OriGene Technologies, MD, USA), or rabbit anti-c-Fos monoclonal antibody (diluted 1:100; code MA5-15055; ThermoFisher Scientific, IL, USA), 0.3% Triton X-100, and 1% normal goat serum for 48 h at 4 °C. After washing in 0.02 M PBS (three times for 10 min each), the sections were incubated in a solution containing goat biotinylated anti-mouse (1:200, Vector, Burlingame, CA, BA-9200, USA) or goat biotinylated anti-rabbit (1:200, Vector, Burlingame, CA, BA-1000, USA), 1% normal goat serum and 0.3% Triton X-100 in 0.02M PBS, pH 7.4 for 60 min at room temperature. After washing in 0.02 M PBS (three times for 10 min each), the sections were placed in the avidin–biotin complex (ABC kit Vectastain, PK-6100, Vector Laboratories, Burlingame, CA, USA) for 45 min and, subsequently, treated with 3.3’-diaminobenzidine (DAB kit, SK-4100, Vector Laboratories, Burlingame, CA, USA). The reaction was stopped by washing the sections in 0.02 M PBS. Sections were then mounted onto gelatin-coated slides, dried overnight at 37 °C, dehydrated in ascending alcohols to xylene, and coverslipped with permanent mounting medium (Entellan; Merck, Darmstaldt, Germany).

### 2.4. Thionin Staining

To facilitate identification of the boundaries of the prefrontal cortex, paraventricular nucleus, and nucleus of the solitary tract, sections adjacent to immunoperoxidase sections were mounted on gelatin-coated slides, dried overnight at 37 °C, defatted 1 h in a mixture of chloroform/ethanol 100% (1:1), rehydrate through descending alcohols to distilled water and stained 30 s in a 0.125% thionin (Fisher Scientific) solution. Subsequently, the sections were dehydrated in ascending alcohols to xylene and cover slipped with permanent mounting medium (Entellan; Merck, Darmstaldt, Germany).

### 2.5. Analysis of Sections

Preparations were observed using a Leica DMRB microscope. Brightfield images were recorded with a Polaroid DMC digital camera (Polaroid Corporation, Cambridge, MA, USA) and DMC 2 software (Polaroid Corporation, Cambridge, MA, USA). Contrast and brightness adjustments were made using Adobe Photoshop CS3 Extended 10.0 software (Adobe Systems, San Jose, CA, USA).

To obtain the density of nNOS-IR neurons (in the prefrontal cortex), oxytocin-IR neurons (in the paraventricular nucleus), and c-Fos-IR neurons (in the nucleus of the solitary tract), immunolabelled cell bodies were plotted by means of a computer-aided digitizing system (Accustage 5.1, St. Shoreview, MN, USA) in a representative series of coronal sections (every fifth section throughout each area). A stereo microscope equipped with drawing tube were used to define the boundaries of the prefrontal cortex, paraventricular nucleus, and nucleus of the solitary tract using adjacent thionin-stained sections. The outlines were superimposed on computer generating plots using Corel Draw X3 (Corel corporation, Ottawa, Ontario, Canada). The area measurements were done from the line drawings by using AxioVision Rel.4.8 (Zeiss, Oberkochen, Germany). The density of immunostained neurons was calculated as number of cell bodies/mm^2^ in each section separately. For each animal, 15 sections were analyzed. Counts were performed separately for each hemisphere and results were averaged between hemispheres. The neuronal counts are expressed as the mean number of somata/mm^2^ ± standard deviation and the data from control diet, milk diet and wean diet groups were compared. The one-way ANOVA test was used for comparing the statistical differences between groups, with a significance level at *p* < 0.05.

AxioVision Rel.4.8 software (Zeiss) was utilized for morphometric analysis of the oxytocin-IR neurons located in the paraventricular nucleus. For each animal, the perikaryal areas of oxytocin-IR cell bodies of 10 nonconsecutive sections were measured after manual tracing of the cell bodies outline. A total of 1000 somata were measured for each group. Data were expressed as mean ± standard deviation (SD). The intensity of immunoreactivity for the oxytocin was analyzed with a semi-quantitative method using Leica DMRB microscope. The intensity was rated as + + + high, + + medium, + low, − absent.

The percentage of the image covered by MOG immunoreactivity were obtained using the automatic threshold algorithm of ImageJ (version IJ 1.46r downloaded from http://imagej.nih.gov/ij/download.html). An adjacent series of sections stained with thionin was used to determine the cytoarchitectonic boundaries nucleus of the solitary tract. For this analysis images were taken using the Leica DMRB microscope under identical acquisition parameters for groups 1, 2, and 3. For each animal, 10 sections were analyzed. All data are given as mean ± standard deviation (SD) and differences between the control diet, milk diet, and wean diet groups were evaluated using one-way Analysis of Variance (ANOVA) test, with a significance level at *p* < 0.05.

## 3. Results

### 3.1. Paraventricular Nucleus: Oxytocin Immunoreactivity

First, we evaluated the morphological and morphometrical characteristics of the oxytocin-immunoreactive neurons, as well as their intensity of immunostaining. Subsequently, we verified the density of the different neuronal populations. Neuronal labeling was somatodendritic and immunoreactivity in particular was cytoplasmic. Neurons with different morphological characteristics and staining intensities (Figure 1A–C; Table 1) were found in the paraventricular nucleus. In some neurons, immunoreactivity for oxytocin could be detected only in the cell body. The most frequently observed neurons were those with polygonal-shaped somata (Figure 1A). However, there were also neurons with fusiform (Figure 1B) or spherical (Figure 1C) cell bodies. The mean perikaryal area was significantly larger in milk diet group than in the control diet and wean diet groups (Figure 1D). In particular, the milk diet group showed the largest areas of polygonal (Figure 1E) and spheroidal (Figure 1G) neurons. 

The intensity of immunostaining was lower in wean diet group than in the control and milk diet groups (Figure 2A–C; Table 1). Comparison among the three groups showed no significant differences in neuronal densities (Figure 2A–G).

### 3.2. Prefrontal Cortex: Myelin Oligodendrocyte Glycoprotein (MOG) and nNOS Immunoreactivity

The percentage of the image covered by MOG immunoreactivity was significantly higher in wean diet group than in control group (Figure 3A–C).

Neurons immunoreactive for nNOS (nitrergic neurons) were distributed in the different cortical layers (Figure 4A–C). Comparison among the three groups showed no difference in nitrergic neuronal densities (Figure 4D). The most frequently observed non-pyramidal neurons were those with angular-shaped somata (Figure 4E,F). There were also non-pyramidal neurons with fusiform cell bodies of different dimensions (Figure 4G).

### 3.3. Nucleus of the Solitary Tract (A2 group): c-Fos Immunoreactivity

The c-Fos immunoreactivity was similar between the control diet, milk diet, and wean diet groups (Figure 5A–G).

## 4. Discussion

Stressors are stimuli that challenge bodily homeostasis and well-being. Accordingly, it is well established that stress is a major risk factor for behavioral disorders. Signals generated by stressors may also arise from gastrointestinal signals. Consequently, in the present study, the neurochemical effects that stress related to the intake of different diets induce on the bovine prefrontal cortex, paraventricular nucleus, and nucleus of the solitary tract has been evaluated.

Previous studies have demonstrated that stressors are able to induce changes on glial populations of the prefrontal cortex, a region involved in many cognitive and emotional behaviors. In particular, early stressful experience in juvenile animals results in defective developmental myelination in the prefrontal cortex. In addition, impaired myelination in the human prefrontal cortex has been reported in a wide range of mood disorders including anxiety and depression [27]. In the present study, we found that in the different groups there are also some differences in MOG immunoreactivity in the prefrontal cortex. In particular, the level of immunoreactivity is significantly higher in the wean diet group than in the control diet group, indicating that myelinating oligodendrocytes in the prefrontal cortex respond to stressors arising from the gastrointestinal tract. Accordingly, ipomyelination in the prefrontal cortex can be induced in human prefrontal cortex by negative experiences inducing chronic stress [28]. Stressors can also affect the neuronal population of the prefrontal cortex, in particular that producing nitric oxide [5,29]. Our data suggest that a different kind of diet does not induce changes in nitric oxide-mediated transmission in the prefrontal cortex.

Stress is characterized by physiological changes, comprising a cascade of neuroendocrine events, in response to stressors. Oxytocin in the paraventricular nucleus modulates physiological and behavioral processes in many vertebrates. In particular, oxytocin shows stress protective and anxiolytic effects and is up-regulated in response to chronic restraint stress. Consequently, oxytocin is an important regulator of stress response, with the capacity to buffer against stressors [12,13,14]. We examined whether diet affects the number of oxytocin-IR neurons in the paraventricular nucleus by comparing three groups of animals. Density analyses indicated that there were no differences among the different groups. In addition, the proportion of different cell types (polygonal, fusiform, and spheroidal) did not vary in the different animal groups either. Interestingly, the intensity of immunostaining for oxytocin was lower in wean diet group than in the control and milk diet groups. Since the intensity of immunostaining could be related to the quantity of oxytocin inside the neurons, we could deduce that oxytocin activity is lowest in wean diet group. These results suggest that in the control and milk diet groups, stress related to different diets increases oxytocin production to inhibit both stress-related hormones and physiological indicators of stress. In the paraventricular nucleus, oxytocin is expressed in magnocellular and parvocellular neurons [10]. Whereas magnocellular neurons modulate adrenocorticotropic hormone (ACTH) release through the release of oxytocin into portal circulation, the parvocellular neurons can regulate the adrenal cortex activity via autonomic outflow [11,12,13,14]. We also describe the morphometrical characteristics of oxytocin-IR neurons in the paraventricular nucleus. The mean perikaryal area of these neurons in the milk diet group is different to that in the control and wean diet groups. In particular, the milk diet group appears to have neurons with the largest cell body (especially neurons with polygonal and spheroidal morphology). This result indicates that many magnocellular paraventricular nucleus neurons in milk diet group are involved in oxytocin secretion, suggesting that in this group a high level of stressors modulate ACTH secretions during different types of stress, releasing oxytocin into portal circulation. These variations indicate that different kinds of diets trigger a change in oxytocin neuronal population. A schematic representation of the role of oxytocin as a stress modulator is reported in Figure 6.

Central signaling pathways arising from the nucleus of the solitary tract (A2 neurons) are essential for regulating stress responses. In fact, A2 neurons selectively target the paraventricular nucleus and increase oxytocin excitability. Our results indicate that neuronal activation in the A2 cell group is not related to different kinds of diet.

## 5. Conclusions

This is the first study investigating the effects of different diets on neurochemistry in ruminants. Our results establish that weaning reduces the expressions of oxytocin in the paraventricular nucleus and stimulates myelination in the prefrontal cortex. This finding supports the importance of maintaining a healthy nutritional lifestyle, similar to that occurring in natural conditions. 

## Figures and Tables

**Figure 1 animals-09-00358-f001:**
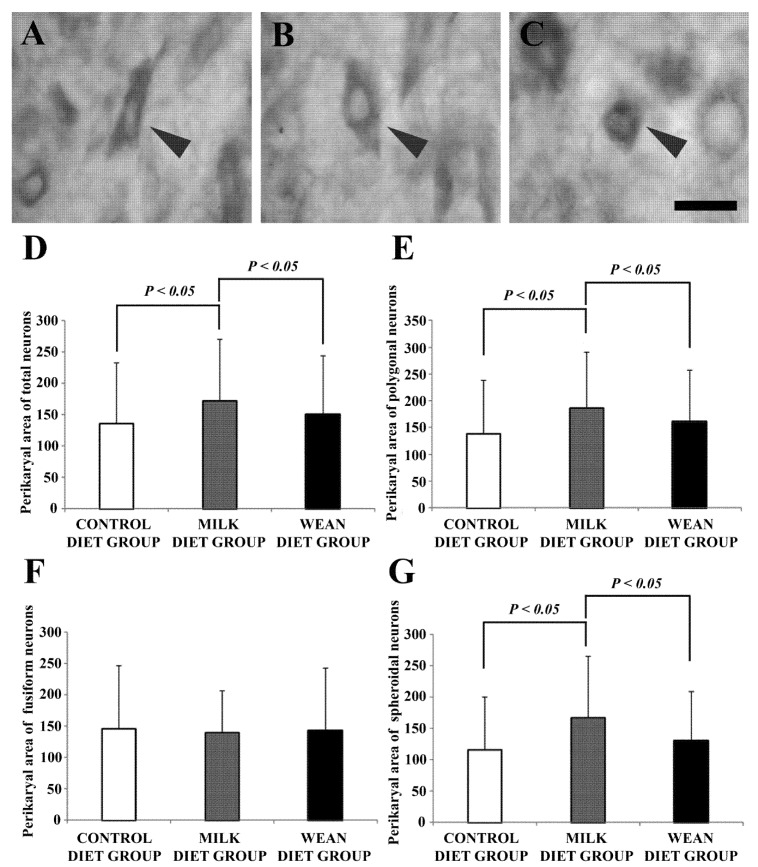
Brightfield photomicrographs of coronal sections (**A**–**C**) and histograms (**D**–**G**) showing morphological types and morphometric analysis of oxytocin-immunoreactive neurons in the paraventricular nucleus. Note that neurons show a polygonal (**A**), fusiform (**B**), and spheroidal (**C**) morphology (arrowheads). The mean perikaryal area is significantly larger in the milk diet group than in the control and wean diet groups (**D**). In particular, the milk diet group exhibits the largest areas of polygonal (**E**) and spheroidal (**G**) neurons. Scale bar = 20 µm in (**C**) (applies to (**A**–**C**)).

**Figure 2 animals-09-00358-f002:**
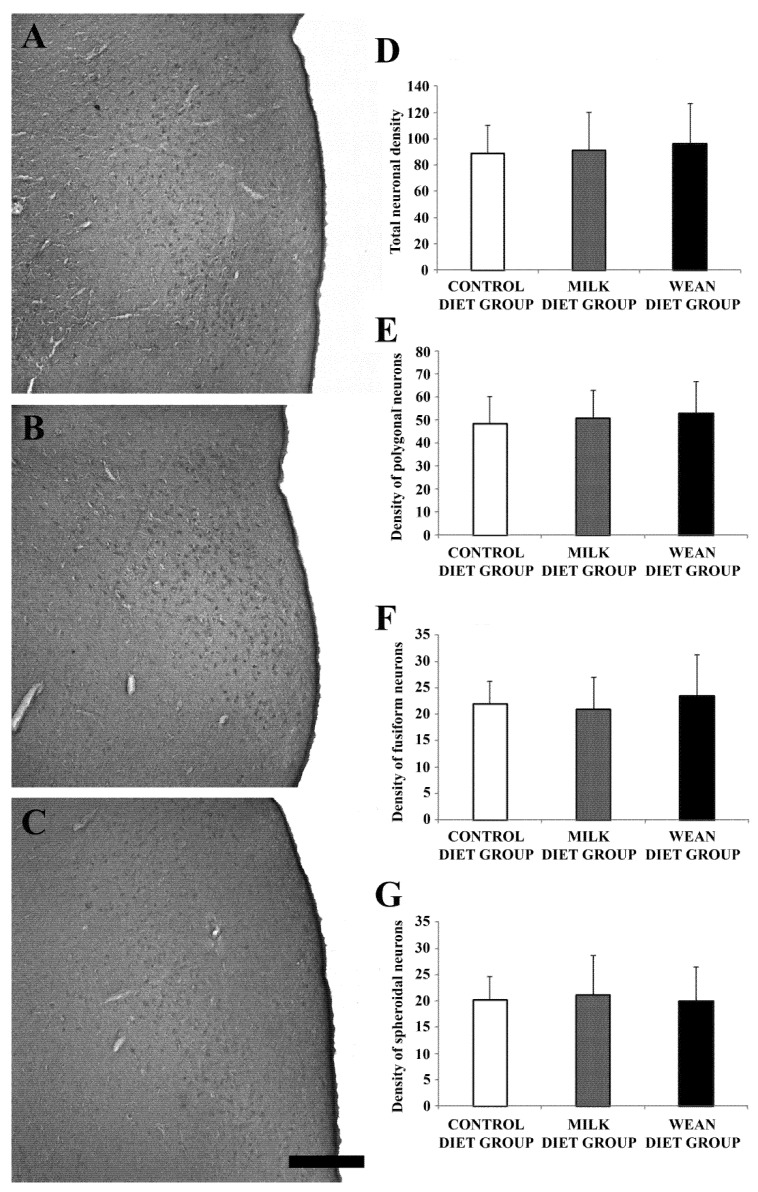
Brightfield photomicrographs of coronal sections (**A**–**C**) and histograms (**D**–**G**) showing the distribution of oxytocin immunoreactivity in the paraventricular nucleus of the control diet group (**A**), milk diet group (**B**), and wean diet group (**C**). The density (number of somata/mm^2^) of oxytocin-immunoreactive (IR) neurons does not vary significantly in the different groups (**D**–**G**). Note that the intensity of immunostaining is lower in wean diet group than in the control and milk diet groups (see also Table 1). Scale bar = 500 µm in (**C**) (applies to (**A**–**C**)).

**Figure 3 animals-09-00358-f003:**
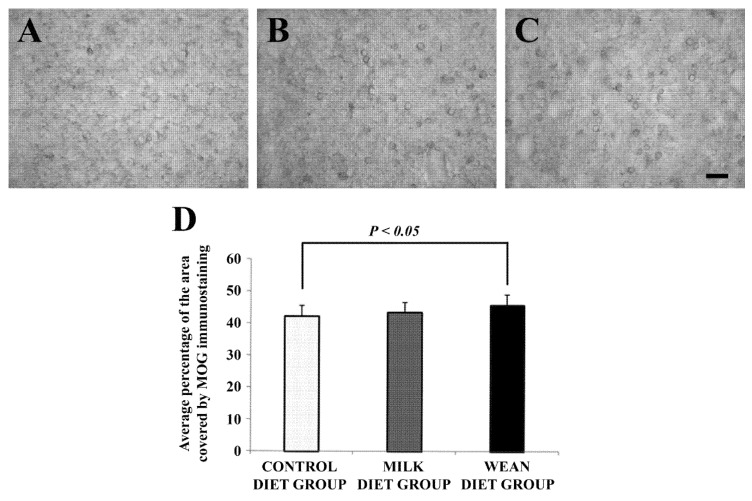
Brightfield photomicrographs of coronal sections (**A**–**C**) and histogram (**D**) showing the distribution of myelin oligodendrocyte glycoprotein (MOG) immunoreactivity in the prefrontal cortex of the control diet group (**A**), milk diet group (**B**), and wean diet group (**C**). Note that the percentage of the image covered by MOG immunoreactivity (**D**) is significantly higher in wean diet group than in control diet group. Scale bar = 20 µm in (**C**) (applies to (**A**–**C**)).

**Figure 4 animals-09-00358-f004:**
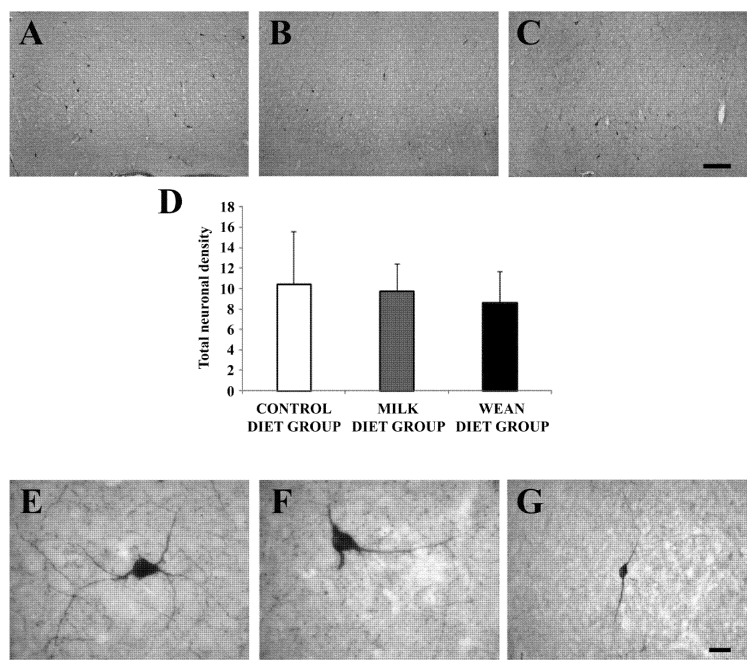
Brightfield photomicrographs of coronal sections (**A**–**C**) and histogram (**D**) showing the distribution of neuronal nitric oxide synthase (nNOS) immunoreactivity in the prefrontal cortex of the control diet group (**A**), milk diet group (**B**), and wean diet group (**C**). Comparison among the three groups shows no significant difference in neuronal densities (number of somata/mm^2^; **A–D**). In every group, nitrergic cells are non-pyramidal neurons with angular-(**E**,**F**) or fusiform-(**G**) shaped cell body. Scale bars = 200 µm in (**C**) (applies to (**A**–**C**)); 20 µm in (**G**) (applies to (**E**–**G**)).

**Figure 5 animals-09-00358-f005:**
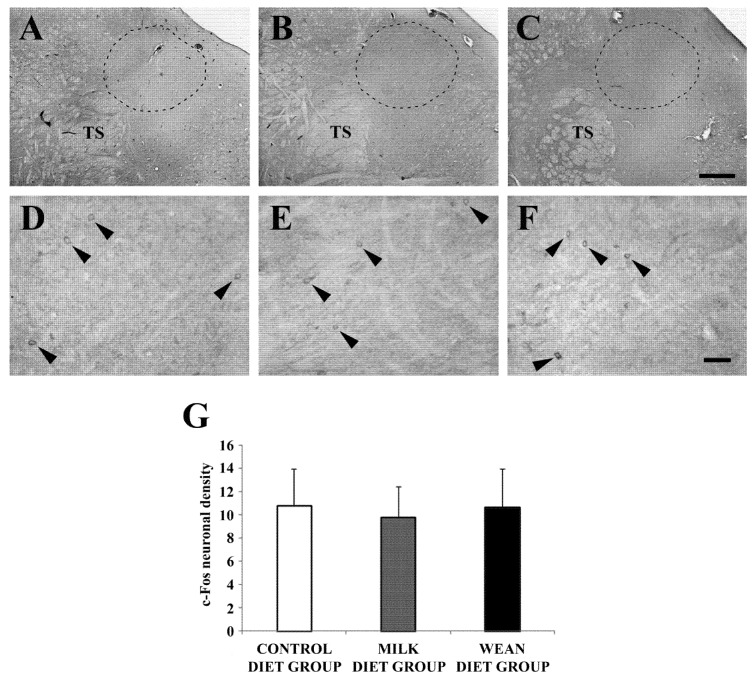
Brightfield photomicrographs of coronal sections (**A**–**F**) and histograms (**G**) showing the distribution of c-Fos immunoreactivity in the nucleus of the solitary tract (bounded by the dotted line) (A2 group) of the control diet group (**A**,**D**), milk diet group (**B**,**E**), and wean diet group (**C**,**F**). The density (number of somata/mm^2^) of c-Fos-immunoreactive neurons is very similar in the different groups (arrowheads). Scale bars = 500 µm in (**C**) (applies to (**A**–**C**)); 20 µm in (**F**) (applies to (**D**–**F**)). Abbreviations: TS, *tractus solitarius*.

**Figure 6 animals-09-00358-f006:**
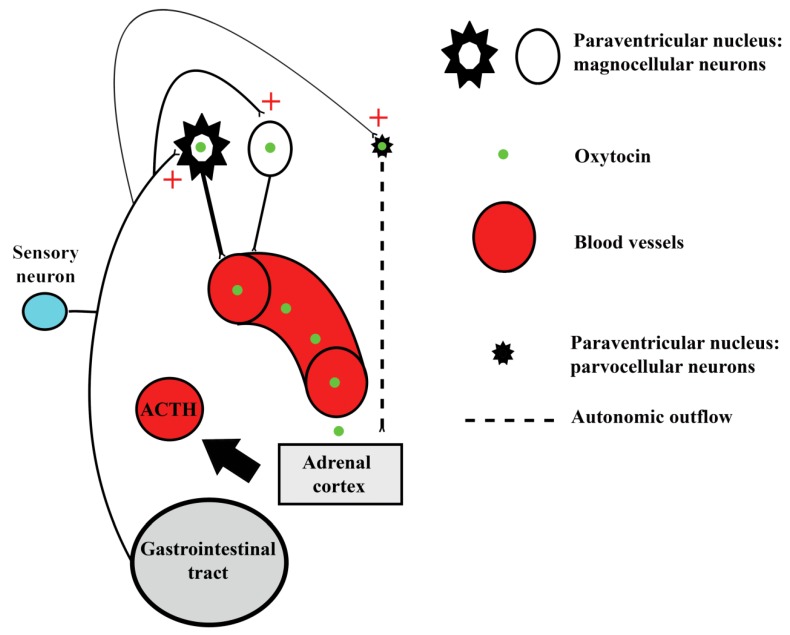
Oxytocin as a regulator of stress response: schematic representation. High level of stressors arising from the gastrointestinal tract increase oxytocin production in magnocellular and parvocellular neurons of the paraventricular nucleus. Magnocellular neurons modulate the adrenocorticotropic hormone (ACTH) activity through the release of oxytocin into portal circulation, whereas parvocellular neurons regulate the adrenal cortex activity via autonomic outflow.

**Table 1 animals-09-00358-t001:** Intensity of immunoreactivity (INI) for the oxytocin in the paraventricular nucleus.

GROUPS	INI
Control diet	+ + +
Milk diet	+ + +
Wean diet	+

The intensity of the staining is expressed as + + + high, + + medium, + low.

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
