# Peer review of "Environment and Behavior: Neurochemical Effects of Different Diets in the Calf Brain"

_animals, 2019, doi:10.3390/ani9060358_

Round 1

Reviewer 1 Report

a. The t-test is not the appropriate statistical test for determining differences between groups.  A one-way ANOVA would be the appropriate method.

b. check the scale bar in Figure 3 it seems larger than 20 micrometers.

c. We need units for perikaryal area for the graphs in figure 1.  What is the relationship of perikaryal area to the conclusions of the paper? And is this relationship described in the manuscript?

d. The standard deviation bars are pretty large, have you considered using standard error of the mean (SEM)?  SEM accounts for the number of samples and typically shows less variation.

e. For figure 2, you state that group 3 had the lowest oxytocin immunostaining, but was it significantly lower?

So I can accept that group 3 showed significantly less oxytocin immunoreactivity compared to group 1 and 2 suggesting less oxytocin activity.  And I can accept group 3 showed significantly more myelin (MOG) immunoreactivity suggesting more myelination activity is occurring.  

  And so you reference that impaired myelination occurs in human disease such as anxiety.  Now don't the results show that the weaned group 3 calves had more myelination activity, not less?  

     And lastly, since you showed differences between the weaned calves and the other groups I think you will have to take a stand and indicate which diet is the most optimal for oxytocin and myelination development.  Really, I think the data shows that weaning reduces oxytocin, suggesting less stress protection; and promotes more myelination.  You could simply make the conclusion that weaning reduces oxytocin compared to the standard commercial diet.

Author Response

REVIEWER 1

a. The t-test is not the appropriate statistical test for determining differences between groups.  A one-way ANOVA would be the appropriate method.

Response: We agree with the Reviewer. The t-test has been replaced by one-way ANOVA test. However, the statistically significant differences remained the same (materials and methods, page 5, lines 193 and 209).

b. check the scale bar in Figure 3 it seems larger than 20 micrometers.

Response: We checked the scale bar of figure 3. Twenty micrometers are the correct value.

c. We need units for perikaryal area for the graphs in figure 1.  What is the relationship of perikaryal area to the conclusions of the paper? And is this relationship described in the manuscript?

Response: In the introduction (lines 61-72) we reported that: ”In the paraventricular nucleus, magnocellular and parvocellular neurons can be identified and classified on the basis of their neurochemical profile (8, 9). Generally, both magnocellular and parvocellular neurons can synthesize oxytocin (10). Magnocellular neurons, through the hypothalamo-neurohypophysal system, release the oxytocin directly into the systemic circulation, modulating different physiological activities such as labor, parturition, lactation, maternal behavior, appetite regulation, gastric distension, blood pressure and electrolyte balance (11). In addition, these cells, releasing the oxytocin into portal circulation, modulate ACTH release, and should be considered possible modulators of stress response (12-14). Parvocellular neurons that secrete oxytocin, on the other hand, projecting to brainstem and spinal cord autonomic neurons, integrate sympathetic/parasympathetic outflow and regulate the activity of the adrenal cortex through an autonomic system (11)”.

In the discussion (lines 350-358) we reported that: ”We also describe the morphometrical characteristics of oxytocin-IR neurons in the paraventricular nucleus. The mean perikaryal area of these neurons in group 2 is different to that in groups 1 and 3. In particular group 2 appears to have neurons with the largest cell body (especially neurons with polygonal and spheroidal morphology). This result indicates that many magnocellular paraventricular nucleus neurons in group 2 are involved in oxytocin secretion, suggesting that in this group a high level of stressors modulate ACTH secretions during different types of stress, releasing oxytocin into portal circulation. These variations indicate that different kinds of diets trigger a change in oxytocin neuronal population.”

d. The standard deviation bars are pretty large, have you considered using standard error of the mean (SEM)?  SEM accounts for the number of samples and typically shows less variation.

Response: Because in figures 5 and, especially, 3 the SD is quite low. If we adopt SEM, bars in figure 3 and 5 would be to short.

e. For figure 2, you state that group 3 had the lowest oxytocin immunostaining, but was it significantly lower?

Response: The intensity of immunoreactivity for the oxytocin was analyzed only with a semi-quantitative method. Consequently, we cannot say that there are statistically significant differences.

f) So I can accept that group 3 showed significantly less oxytocin immunoreactivity compared to group 1 and 2 suggesting less oxytocin activity.  And I can accept group 3 showed significantly more myelin (MOG) immunoreactivity suggesting more myelination activity is occurring. And so you reference that impaired myelination occurs in human disease such as anxiety.  Now don't the results show that the weaned group 3 calves had more myelination activity, not less? 

Response:  As reported in the conclusions, our results establish that weaning (group 3 of animals) reduces the expressions of oxytocin in the paraventricular nucleus and stimulates myelination in the prefrontal cortex.

g) And lastly, since you showed differences between the weaned calves and the other groups I think you will have to take a stand and indicate which diet is the most optimal for oxytocin and myelination development.  Really, I think the data shows that weaning reduces oxytocin, suggesting less stress protection; and promotes more myelination.  You could simply make the conclusion that weaning reduces oxytocin compared to the standard commercial diet.

Response: As reported in figure 3, the percentage of the image covered by MOG immunoreactivity is significantly higher in wean diet group than in control diet Group.

Reviewer 2 Report

The article entitled “Environment and behavior: neurochemical effects of different diets in 
the calf brain
is an interesting article in which authors used immuno-histochemistry approaches as major tools to test their hypothesis.  In this study, the authors determined that weaning diet reduces the expression of oxytocin in Paraventricular Nucleus (PVN) and that leads to the stimulation of myelination in prefrontal cortex (PFC) which is very interesting. However, utilizing immunoelectron microscopy technology to assess the myelination in PFC regions would be great way to validate such study, if the authors have possibility to exploit electron microscopy that would really makes this study to understand in greater depth and make it  more impactful. Still, the authors need to mention clearly which layers are specifically targeted for myelination within the PFC.

Minor issues:

Line 29-30: Please remove all “Group 1”, “Group 2” “Group,3” information and add actual experimental group. Such as Control, milk diet and wean diet. Group 1, 2, 3 are additional information and you really can substitute it throughout manuscript including figures. Please make sure that conclusion part of you abstract is consistent with the manuscript “Conclusion “ that you provided at end of the manuscript

Line 37-96: In the “Introduction” section please make sure that the presentation style in the introduction section is consistent or matching with abstract section. Currently there looks some misalignment. For example:  In opening sentences of an abstract, you have highlighted about “Stress response”.  However in the opening sentences in the “Introduction “section, you have opening sentence about calves feeding habit”. I would suggest to either amending your abstract or introduction making both consistent with each other.

Line 83-86: Any paragraph that consists of less than 3 sentences are usually better to merge in other paragraph in relevant place rather than forming a new small paragraph separately.

Line 112: “ pen” wording can be substituted or defined clearly.

Line 114-119: It would be more scientifically sound if you could provide dose/meal/body weight of calves if possible.

Line 128: “4 %  paraformaldehyde 0.1M Sodium phosphate buffer”. This wording does not explain whether such substances are commercially available or you dissolved relevant counterparts to get it.

Line 130: For how many days 30 % sucrose solution are used during incubation at 4 oC?

Line 213-268: In all the representative quantification, it is better to replace “Group 1”, “Group 2” and “Group 3” with original intervention so that one should not need to toggle between methods and results as I mentioned earlier.

-Please provide sample size in each and every quantification bar or in their legend (usually p-value and n should come together).

-Did you look at Supraoptic nucleus as they also produce oxytocin? Since it is diet related study, it would still be relevant to raise the insulin part in connection with oxytocin activation in your discussion.

Major:

Line 10-32:   “Simple Summary” can be merged to the “Abstracts” section. No need to provide separate subheading for “Simple Summary”. If needed some points from these sections should be removed and added to the “Introduction” section.

-Why did not you choose to show the image in higher magnification such as Confocal microscopy? This could have given deeper understanding in morphology of the cell with better quality picture.

-In your result section, please try to provide the rationale or purpose of generating each figure in support your hypothesis? What are the sequence of execution of the experiments and how this different strategy validates your study to reach to the same conclusions in logical fashion? That should be properly explained preferably in your “Results” section.  For this, you can explain “First, we examine …..”

Second having understood the basic “we next assess this issue because first was not enough to provide ……”

“This technique offers better than first approach and they are more specific”. Such a logical transition of the text will make reader better follow the manuscript.

Line 270-316: More citation is needed to show consistency and discrepancy of your findings with other author’s work. Currently, in your “Introduction” or “Discussion” the human relevance of such study is not mentioned. Please provide the limitations or caveats of the study as well as future direction in the field.

-At the end of your “Discussion” it will be good to provide a figure showing hypothetical sketch/model for projection of PVN neurons in the brain and their pathway. By doing that you could easily marked your findings in summary form so that the finding that you brought out in this study could be visualize in a simple schematic sketch (You may see similar sketch in Eliava et al. 2016, Neuron; but you should make it relevance to your own study). In summary, the present “discussion” section is relatively short and adopting this approach will make your points more insightful from different angles.

Author Response

REVIEWER 2

The article entitled “Environment and behavior: neurochemical effects of different diets in the calf brain” is an interesting article in which authors used immuno-histochemistry approaches as major tools to test their hypothesis.  In this study, the authors determined that weaning diet reduces the expression of oxytocin in Paraventricular Nucleus (PVN) and that leads to the stimulation of myelination in prefrontal cortex (PFC) which is very interesting. However, utilizing immunoelectron microscopy technology to assess the myelination in PFC regions would be great way to validate such study, if the authors have possibility to exploit electron microscopy that would really makes this study to understand in greater depth and make it  more impactful. Still, the authors need to mention clearly which layers are specifically targeted for myelination within the PFC.

Response: We agree with the Reviewer, the immunoelectron microscopy is a good technology to study the myelination in prefrontal cortex, but we don't have this technology at our disposal.

Minor issues:

Line 29-30: Please remove all “Group 1”, “Group 2” “Group,3” information and add actual experimental group. Such as Control, milk diet and wean diet. Group 1, 2, 3 are additional information and you really can substitute it throughout manuscript including figures. Please make sure that conclusion part of you abstract is consistent with the manuscript “Conclusion “ that you provided at end of the manuscript

Response: We have done as requested by the Reviewer. Consequently, we have removed “Group 1”, “Group 2” and “Group3” and added control diet group, milk diet group wean diet group.

Line 37-96: In the “Introduction” section please make sure that the presentation style in the introduction section is consistent or matching with abstract section. Currently there looks some misalignment. For example:  In opening sentences of an abstract, you have highlighted about “Stress response”.  However in the opening sentences in the “Introduction “section, you have opening sentence about calves feeding habit”. I would suggest to either amending your abstract or introduction making both consistent with each other.

Response: The opening sentences of the abstract (line 20, page 1) have been changed: “Calves reared for the production of white veal are subjected to stressful events due to the type of liquid diet.”

Line 83-86: Any paragraph that consists of less than 3 sentences are usually better to merge in other paragraph in relevant place rather than forming a new small paragraph separately.

Response:  Sorry, but we do not understand this answer.

Line 112: “ pen” wording can be substituted or defined clearly.

Response: Pen is a small enclosure for livestock. In the 2.1. Animals and ethical statement line 114, pag 3, we added: “…….pen (a small enclosure for livestock) and….

Line 114-119: It would be more scientifically sound if you could provide dose/meal/body weight of calves if possible.

Response: At the end of 2.1. Animals and ethical statement, line 121-125, pag.3 we have added:

At the beginning of the research the average weight of the calves was as follows: 49.40 kg for the control diet group; 49.60 kg for the milk diet group; 49.91 kg for the wean diet group. At the end of the housing period, after 183 days, the average weight of the animals was 278.88 kg for the control diet group, 232.38 kg for the milk diet group and 213.88 kg for the wean diet group.

Line 128: “4 %  paraformaldehyde 0.1M Sodium phosphate buffer”. This wording does not explain whether such substances are commercially available or you dissolved relevant counterparts to get it.

Response: The 4 %  paraformaldehyde 0.1M Sodium phosphate buffer is prepared immediately before the use.

Line 130: For how many days 30 % sucrose solution are used during incubation at 4 oC?

Response: Until tissue sinks (about one week)

 Line      213-268: In all the representative quantification, it is better to replace      “Group 1”, “Group 2” and “Group 3” with original intervention so that one      should not need to toggle between methods and results as I mentioned      earlier.

Response: We have done as requested by the Reviewer.

Please provide sample size in each and every quantification bar or in their legend (usually p-value and n should come together).

Response: Each figure legend has the value of the scale bar

Did you look at Supraoptic nucleus as they also produce oxytocin? Since it is diet related study, it would still be relevant to raise the insulin part in connection with oxytocin activation in your discussion.

Response:  The sopraoptic nucleus was not deeply observed. However, some preliminary observations indicate that also in this area the immunoreactivity for the oxytocin is lower in weaned diet than in other two groups.

Major:

Line 10-32: “Simple Summary” can be merged to the “Abstracts” section. No need to provide separate subheading for “Simple Summary”. If needed some points from these sections should be removed and added to the “Introduction” section.

Response: Animals article requires the summary separate from the abstract.

Why did not you choose to show the image in higher magnification such as Confocal microscopy? This could have given deeper understanding in morphology of the cell with better quality picture.

Response: We have not the possibility to use confocal microscopy.

In your result section, please try to provide the rationale or purpose of generating each figure in support your hypothesis? What are the sequence of execution of the experiments and how this different strategy validates your study to reach to the same conclusions in logical fashion? That should be properly explained preferably in your “Results” section.  For this, you can explain “First, we examine …..” Second having understood the basic “we next assess this issue because first was not enough to provide ……”“This technique offers better than first approach and they are more specific”. Such a logical transition of the text will make reader better follow the manuscript.

Response: In the 3.1. Paraventricular nucleus: oxytocin immunoreactivity we added (line 213-215, pag 5): “First, we evaluated the morphological and morphometrical characteristics of the oxytocin-immunoreactive neurons, as well as their intensity of immunostaining. Subsequently, we verified the density of the different neuronal populations.”

Line 270-316: More citation is needed to show consistency and discrepancy of your findings with other author’s work. Currently, in your “Introduction” or “Discussion” the human relevance of such study is not mentioned. Please provide the limitations or caveats of the study as well as future direction in the field.

Response: In the discussion (lines 328-329, pag. 12) we added: ”Accordingly, changes of myelination in the prefrontal cortex can be induced in human prefrontal cortex by negative experiences inducing chronic stress (28)”. There is a new reference (28) Nickel, M.; Gu, C. Regulation of Central Nervous System Myelination in Higher Brain Functions. Neural Plast. 2018, 5, 1-12.

At the end of your “Discussion” it will be good to provide a figure showing hypothetical sketch/model for projection of PVN neurons in the brain and their pathway. By doing that you could easily marked your findings in summary form so that the finding that you brought out in this study could be visualize in a simple schematic sketch (You may see similar sketch in Eliava et al. 2016, Neuron; but you should make it relevance to your own study). In summary, the present “discussion” section is relatively short and adopting this approach will make your points more insightful from different angles.

Response: We added the figure 6 concerning the role of oxytocin as a stress modulator.

Round 2

Reviewer 1 Report

Well done revisions, the authors have fixed all the points of concern I had and have very appropriately revised the manuscript.  The present work is now acceptable and constitutes an interesting and intriguing article.

This manuscript is a resubmission of an earlier submission. The following is a list of the peer review reports and author responses from that submission.